POINT OF VIEW

# Europe's first and last field trial of gene-edited plants?

**Abstract** On 5 June this year the first field trial of a CRISPR-Cas-9 gene-edited crop began at Rothamsted Research in the UK, having been approved by the UK Department for Environment, Food & Rural Affairs. However, in late July 2018, after the trial had started, the European Court of Justice ruled that techniques such as gene editing fall within the European Union's 2001 GMO directive, meaning that our gene-edited Camelina plants should be considered as genetically modified (GM). Here we describe our experience of running this trial and the legal transformation of our plants. We also consider the future of European plant research using gene-editing techniques, which now fall under the burden of GM regulation, and how this will likely impede translation of publicly funded basic research.
DOI: https://doi.org/10.7554/eLife.42379.001

**JEAN-DENIS FAURE AND JOHNATHAN A NAPIER***

*For correspondence: johnathan.napier@rothamsted.ac.uk

**Competing interests:** The authors declare that no competing interests exist.

## Introduction

Agricultural production of oilseeds is steadily increasing due to the high demand for food, animal feed and new industrial uses (*USDA, 2018*). Innovation is required to meet market needs, and also to facilitate a more sustainable use of agricultural inputs, such as chemicals, fertilizers and irrigation. To help address the challenges all of this represents, we have been exploring the genetic potential of the oilseed *Camelina sativa*. This plant is native to Southeast Europe and Southwest Asia and has an ancient history of culture, but was largely replaced by oilseed rape (canola). However, in recent years, Camelina was rediscovered as a promising crop due to its resilience, relatively low agricultural input requirements, and resistance to biotic and abiotic stresses (*Guy and Ehrensing, 2008*). Camelina also presents an interesting oil profile for food, feed and bio-based uses; it contains more than 50% polyunsaturated fatty acids (35% α-linolenic acid) alongside tocopherols (Vitamin E), which enhance oil stability (*Nguyen et al., 2013*). Finally, Camelina is an excellent translational system to validate new traits, since its genome is closely related to the model plant *Arabidopsis thaliana* (*Kagale et al., 2014*). Equally, ease of genetic modification (*Lu and Kang, 2008*) allows rapid development of new traits by conventional as well as biotechnological modification either by transgenesis or gene editing.

One specific trait of interest in oilseeds, including Camelina, is increased levels of oleic acid, a versatile and valuable monounsaturated fatty acid with high oxidative stability (*Vanhercke et al., 2013*). The market demand for oleic acid-rich oils continues to grow in both North America and Europe, predominantly as a result of efforts to reduce the level of saturated fatty acids in processed food (*Watson, 2018*). Several strategies have been used to increase the oleic acid content of Camelina seed oil, including ethyl methanesulfonate (EMS)-mutagenesis, suppression via RNA interference (RNAi) and, more recently, CRISPR-Cas9 gene editing (*Kang et al., 2011*; *Nguyen et al., 2013*; *Morineau et al., 2017*; *Jiang et al., 2017*). A clear target for modulation is the FAD2 Δ12-desaturase, which is the primary enzyme responsible for the conversion of monounsaturated oleic acid to diunsaturated linoleic acid. Gene editing of *FAD2* in the hexaploid Camelina is particularly advantageous as it enables the fine-tuning of oleic acid accumulation by selecting appropriate allelic combinations within the three *FAD2* homeologues present in the three sub-genomes (*Kang et al., 2011*).

Targeted disruption by CRISPR-Cas9 (programmed with two guide RNAs which recognised sequences conserved in all three *FAD2* homeologues) resulted in a population of Camelina plants in which one or more *FAD2* DNA sequences were mutated. The triple knock-out mutant plants (designated *fad2 -/-/-* to indicate that all three homeologues were disrupted) had the highest level of oleic acid and a developmental phenotype with slow growth, twisted leaves and delayed bolting when grown under glasshouse conditions (*Morineau et al., 2017*). This suggested that a reduction in polyunsaturated fatty acids within the vegetative tissues could impact agronomic performance. This is in part due to the expression of *FAD2* in both seed and vegetative tissues and the disruption of this in the GE lines. However, only field trials could provide a real assessment of the relationship between altered oleic acid levels and efficient agronomic performance in these GE mutants.

To be able to study the plants in agriculturally relevant conditions, we decided to proceed toward field trial evaluation of these oleic-accumulating GE Camelina lines. We compared wild-type Camelina with two GE lines: the triple *fad2-/-/-* line (designated A7) and a second line (F4-24) in which two of the three *FAD2* homeologues are disrupted (*fad2-/-/+*). Unlike line A7, the allelic combination in F4-24 did not perturb the overall phenotype. In addition, we wished to

obtain clarity and precedent as to the regulatory status of GE plants undergoing experimental environmental release and field evaluation. Rothamsted Research has carried out successive GM Camelina field trials since 2014, therefore we set out to determine what official approvals would be needed for a field trial of GE Camelina in the UK, opening the process up for scientific learning and public scrutiny. More recently, it became apparent that, while we were conducting our trial, researchers at the Flemish Institute for Biotechnology (VIB) were carrying out a nationally authorised but covert field trial of GE maize in Belgium. As a crucial difference, one goal of our trial was to ensure that the issue of gene editing and regulatory approval was very much in the open and the focus of public discussion.

## Status update

In absence of unambiguous information regarding the regulatory status of GE plants, in November 2017 we asked for clarification from the Department for Environment, Food and Rural Affairs (DEFRA), the UK competent authority in this area. DEFRA sought advice from their independent Advisory Committee on Releases to the Environment (ACRE), asking them to consider a number of points regarding our GE *fad2* Camelina. Specifically, DEFRA sought clarification on: (i) if these GE lines could have been produced by traditional breeding methods; (ii) if

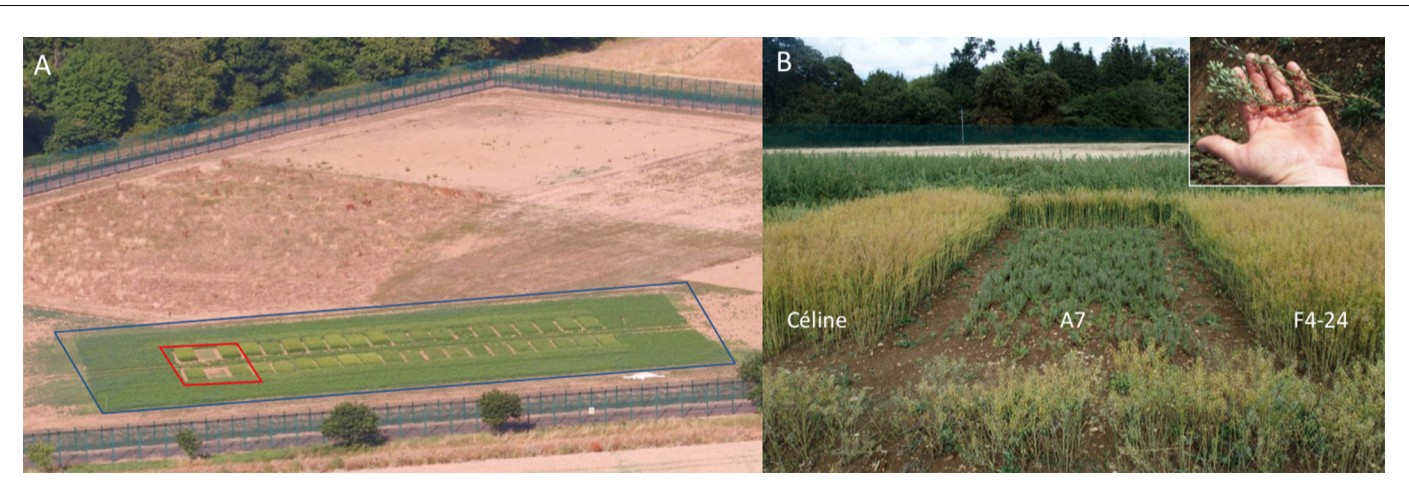

**Figure 1.** Field trial of gene-edited Camelina. (**A**) Aerial view of the trial setting on 12 July 2018, 5 weeks after sowing with two blocks per line (GE lines boxed in red – from left to right, plots are Control Celine, A7 and F4-24). Field irrigation was provided due to the unusually hot and dry UK summer. (**B**) Gene-edited lines A7 and F4-24 with control Céline on 13 August 2018, 9 weeks after sowing (A7 dwarf phenotype in inset).
DOI: https://doi.org/10.7554/eLife.42379.002

they contained any DNA from the CRISPR-Cas9 transformation vector; and (iii) if the CRISPR-Cas9 gene-editing technique was a form of mutagenesis and how recombinant nucleic acid molecules were used in the generation of these plants.

These questions were likely prompted by the Opinion provided by Advocate General Bobek to the European Court of Justice (ECJ) (*Bobek, 2018*) as part of ongoing deliberations as to the status of new breeding techniques such as gene editing. To help address some of these points, we provided additional data as to the presence or absence of the T-DNA insert, based on gene-specific PCR and screening for the visible marker DsRed which was also present in the transgene cassette. ACRE considered these points and also discussed them at an open meeting on 23 March 2018. Their advice to DEFRA, published on 18 May 2018 (*ACRE, 2018*), was that the mutations present in the GE lines could have been produced through traditional breeding techniques. Moreover, although this mutagenesis involved recombinant nucleic acids, these elements (e.g. Cas9 and associated guide RNA) were no longer present in the lines proposed for release. Therefore, ACRE concluded that "it would not be possible to determine whether these lines had been produced by genome-editing or by traditional mutagenesis because they would be genetically indistinguishable". As a result, DEFRA concluded that the Camelina lines were not genetically modified organisms for the purposes of Part VI of the Environmental Protection Act (1990) as applied in England, and as such, there was no requirement to seek consent from the Secretary of State for Environment, Food and Rural Affairs prior to their release. Presciently, DEFRA also noted that this status might change as a consequence of any ruling from the ECJ.

Contemporaneous to the discussion about gene editing above, Rothamsted Research applied to DEFRA on 20 February 2018 for consent to carry out a new GM Camelina field trial. This application, Genetically Modified Organisms: Rothamsted Research (18/R08/01), sought permission to evaluate 14 different GM Camelina lines (predominantly engineered to accumulate omega-3 fish oils – see *Usher et al., 2017*). The application was placed on the DEFRA website for public comment for 48 days from the date of submission. The application was also publicised by advertisement in the national press on 23 February and through a press release from Rothamsted Research. As part of the

description of the trial layout, we included the two GE lines, as they would be grown alongside the GM lines and the wild-type control. The application and any submissions from the public consultation were considered by ACRE at their open meeting on 23 March 2018, and on their advice, consent for this GM trial was also approved on 18 May 2018 (*DEFRA, 2018*). On this basis, the fully authorised release of both GM and GE Camelina could proceed.

## Down on the farm

Seeds were sown in two 9 $m^2$ blocks at a density of 300 per $m^2$ on 5 June 2018. The total area sown was 2128 $m^2$, with GM and GE plants contributing a total of 306 $m^2$. The GM and GE Camelina were surrounded by a 6-metre-wide pollen barrier of wild-type Camelina, with the larger trial site secured with a deer-proof double-fence (*Figure 1A*). As mentioned above, previous glasshouse studies of *fad2 -/-/-* GE mutants had indicated that the resulting high oleic content present in vegetative tissues perturbed development (*Morineau et al., 2017*). Given the importance of polyunsaturated fatty acids in membrane lipid functionality, it is unsurprising that the genetic inactivation of one of the key desaturase enzymes results in phenotypic alterations. However, the phenotype (severely dwarfed plants) observed in the field was more extreme than that previously seen under glasshouse conditions, emphasising the importance of evaluation of plant performance by actual field trials (*Figure 1B*). In this case, it was clear that this particular allelic combination, although generating a high oleic acid seed oil, resulted in very significant growth defects, meaning new allelic combinations should be evaluated for less deleterious vegetative effects.

After sowing in early June, the trial progressed well. Away from the fields, the status of the GE plants was being examined by the ECJ, and 25 July 2018, just as the plants had finished flowering and started to set seed, the ECJ pronounced that new gene-editing technologies such as CRISPR-Cas9 fell under the 2001 GMO directive, meaning that our GE plants were now, in the eyes of the ECJ, metaphorically (but not literally) transformed into GM plants. In line with their previous advice, we contacted DEFRA as to the updated regulatory status of these GE plants. DEFRA confirmed that the plants were now considered GM and they would have to be harvested and destroyed in the same way as the authorised GM lines. Fortunately, due to

experimental design, as opposed to regulatory requirement, the GE lines were being grown within the confines of the 18/R8/01 GM trial, and therefore were managed in exactly the same way as the GM lines. As a result, both trials proceeded to a successful harvest and the GE Camelina lines could be fully assessed for field performance.

## The making of a mutant

In the context of the new European classification of GE plants, one could wonder how their regulatory status will be determined in practise. While the presence or absence of a transgene is easily monitored, GE plants will be only distinguished by the new mutation they carry. Even for a small experimental GM field trial as described here, the risk assessment which forms the basis on which consent is given, focuses on the different foreign DNA elements added to the host [see Genetically Modified Organisms: Rothamsted Research (18/R08/01) for full details]. However, the GE plants under study in this case do not contain any transgenes and, as DEFRA noted in their advice, the mutations in the *FAD2* desaturase could have occurred by traditional mutagenesis, or even natural variation. So what diagnostic technique could be used to find a small deletion within an entire genome, and perhaps more importantly, how might one determine how this occurred? If gene editing was used to recreate a mutation that was already known to exist in nature, how could this process be defined and traced? This conundrum is one that will need to be urgently addressed by the EU's regulatory agencies, not least of all if other countries and trading partners start the commercial cultivation of GE commodity crops and these are imported into Europe.

Strictly speaking, while most GE mutations are indeed induced by the Cas9-mediated sequence-specific cleavage, they were ultimately produced by the plant DNA repair machinery in a relatively random way, replicating "classical" chemical or ionising radiation mutagenesis. In the case described here, the CRISPR-Cas9 generated mutations in *FAD2* displayed variations in the size of the deletion, which lead to the different mutant alleles (*Morineau et al., 2017*). Interestingly, if not confusingly, mutations in the *FAD2* genes of soybean, rapeseed and sunflower have been previously selected to produce high oleic acid cultivars that have been grown for many years. The ECJ ruling stated that *"only organisms obtained by means of techniques/ methods of mutagenesis which conventionally have been used in a number of applications and have a long safety record are excluded from the scope of that [GMO] directive"*. Thus, the high oleic acid Camelina lines described here differ from the other high oleic acid crops only by the mutagenesis process and not by the actual mutations in *FAD2* genes.

One of the virtues of the CRISPR-Cas9 system is the precision by which it introduces a single mutation, which is in stark contrast to conventional breeding or mutagenesis which introduces many. But, in the eyes of the ECJ, the multiple mutations obtained by such techniques are allowable simply on the basis of previous safe use. Obviously, obtaining a "*long safety record*" for growing these GE lines in the context of such strict European regulation would be very difficult, if not impossible. But it is interesting to note that this ECJ ruling might now encompass plants that were generated without gene editing or genetic modification but which do not have a long history of use – this again could impede innovation in the plant breeding sector. Ultimately, the *de facto* reclassification of gene editing as a form of genetic modification presents an enormous burden on researchers (public or private) trying to convert their ideas into innovations and impactful outcomes. Given that the EU has only approved one new GM crop (BASF's Amflora potato) in the last decade (and even that approval was subsequently annulled by General Court of the European Union), the pipeline for translating the skills and know-how of the EU research base in this sector looks blocked, and equally unlikely to attract inward investment. It is hard to believe that this is either a desired or desirable outcome.

## The importance of field trials

Field trials provide a tremendous amount of otherwise unknown information on how plants respond to environmental changes in the field, under agricultural systems. Based on our recent experience (*Usher et al., 2017*), and backed up by a plethora of previous observations, it is vital to move from controlled environment to the field to validate any potentially interesting phenotype. If part of the ambition of basic plant sciences research is to deliver to the needs of agriculture and "feed the world" then there needs to be a greater flux from growth cabinet to field; also, there needs to be an appreciation that performance in the first is no guarantee of something similar in the field. In the case of GE,

such edited plants have the potential to uncover new and extremely useful traits for crops, notably in the context of more sustainable agriculture, where a reduction in pesticides, fertilisation and irrigation is desired. Only field trials can provide the environmental gauntlet to challenge these new traits under realistic conditions and allow for the selection of the most relevant alleles.

In conclusion, the Rothamsted Camelina 2018 GE field trials provided a wealth of essential data and enabled the evaluation of the potential of new trait. Access to field trials are an essential component in the demonstration of efficacy for any new crop trait. Limiting the feasibility of GE field trials by expanding the complexity of the regulatory process and the associated financial burden of dedicated experimental sites will certainly hinder research and limit the contribution EU research can make to meet the challenge of the UN's Sustainable Development Goals. Sadly, there is a general perception that carrying out GM field trials is a burden that is best avoided. However, based on our experiences at Rothamsted Research, not only do such trials provide vital information as to the utility (or not) of a trait (*Bruce et al., 2015*), they also provide a strong focal point for science communication and public engagement. Collectively, we must continue to advance our understanding of plant sciences and crop biology. Therefore, we strongly urge that the scientific community continues to make the case in Europe for access to research field trials for the evaluation of GE and GM crops. This needs to take the form of not just appropriate infrastructure (which can range from something akin to the Swiss Federal Government "Protected Site" at Reckenholz, the facilities at Rothamsted Research or just a suitable field), but also both financial and institutional support to carry out such experiments. Thus, funding agencies should not decline to support such activities, and academic institutions should not shrink from carrying them out. Otherwise, translation and innovation in plant sciences in Europe will be significantly impaired.

## Acknowledgements
We thank Dr Lihua Han for her excellent help in running GM trial 18/R8/01.

**Jean-Denis Faure** is at the Institut Jean-Pierre Bourgin (IJPB), INRA, AgroParisTech, CNRS, Saclay Plant Sciences (SPS), and the Université Paris-Saclay, Versailles, France

**Johnathan A Napier** is in the Department of Plant Sciences, Rothamsted Research, Harpenden, United Kingdom

johnathan.napier@rothamsted.ac.uk

http://orcid.org/0000-0003-3580-3607

*Author contributions:* Jean-Denis Faure, Conceptualization, Resources, Writing—original draft; Johnathan A Napier, Resources, Methodology, Project administration, Writing—review and editing

*Competing interests:* The authors declare that no competing interests exist.

## Funding

| Funder | Grant reference number | Author |
|---|---|---|
| Association Instituts Carnot | 75000058 3BCAR | Jean-Denis Faure |
| AgroParisTech | OLEOCAM | Jean-Denis Faure |
| Institut National de la Recherche Agronomique | FIELDCAM | Jean-Denis Faure |
| Biotechnology and Biological Sciences Research Council | BBS/E/C/000I0420 | Johnathan A Napier |

The funders had no role in study design, data collection and interpretation, or the decision to submit the work for publication.

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
