## [Decision Letter]

Thank you for submitting your article "Europe's first and last field trial of gene-edited plants?" for consideration by *eLife*. Your article has been reviewed by three peer reviewers, and the evaluation has been overseen by Stuart King as an Associate Features Editor and Peter Rodgers as the Features Editor. The following individuals involved in review of your submission have agreed to reveal their identity: Sten Stymne (Reviewer #2).

Following consultation with the reviewers, the Associate Features Editor has drafted this decision to help you prepare a revised submission.

The Features editors may also contact you separately about some editorial issues that you will need to address.

Summary:

This opinion piece describes how the authors' ongoing field trial of genome-edited *Camelina sativa* plants changed legal status following a decision made by the European Court of Justice. Essentially, in a single day, the plants switched from not being regulated by the EU's GM directive to becoming just that. The article gives a personalised response to that decision; it also touches on the rationality of the ruling and its profound effects in the evaluation of crops in Europe.

This is a story worth telling, but the following revisions would strengthen the manuscript.

Essential revisions:

-Refine the aim and take-home message

1) While sympathizing with the authors' frustrations, the reviewers would encourage the authors to think more about what it is they hope to achieve with their article and make that clearer in the text. The article touches on many topics but there is often little in the way of facts that would persuade the readers of the importance of the issues.

2) The reviewers were also puzzled what the take-home message is in the last paragraph. They all agree that field trials are essential to assess a plant's performance, but there was a feeling that this final section seemed slightly self-contradictory in places (for example, "Limiting the feasibility of field trials of GE by the complexity of the regulatory process.… will certainly hinder research" and "And whilst the application for Consent involves a detailed submission, this is no more arduous than that required for a manuscript for publication"). Please consider rephrasing this section to clarify and strengthen its message.

-Introduce more science and context

3) For an unfamiliar reader, it is not clear what advantages genome editing has over conventional mutation techniques, since the authors state that the two approaches can do the same thing. The techniques should be more clearly outlined, and the differences discussed.

4) Information should be provided on the biosynthesis of unsaturated fatty acids and the role of FAD2. This could be added in a Box, like https://elifesciences.org/articles/39233#box1, to avoid disrupting the flow of the main text.

5) It is clear to those working in the field that the EU's GM directive is directed to regulate the technology used in altering the properties of a plant and not the introduced property per se. Therefore, the argument that the authors put forward that traditional mutation breeding and mutations through CRISPR-Cas9 cannot be distinguished is correct, but this does not change the interpretation of the directive. The reviewers felt that the authors could discuss the problems with the GM directive concerning mutation breeding more to make these issues apparent to unfamiliar readers.

6) The reviewers noted that in most cases CRISPR-Cas9 mutations through transgenes produce many off-target mutations and DNA rearrangements. Often pieces of DNA of the vector plasmid are integrated in the genome (which are not detected by PCR of the marker gene or CRISPR-Cas9 genes). Although off-target mutations are much more frequent in plants that have undergone conventional mutation breeding, it cannot be excluded that the authors' *Camelina* contain small pieces of foreign DNA. The authors are advised not to state that their *Camelina* is free from foreign DNA, unless it has been rigorously proved via whole genome sequencing.

7) The authors should also be aware that, in 2017-2018, other field trials with genome-edited plants have been performed (including a trial with maize in Belgium). This and any other examples should be mentioned in the article.

---

## [Author Response]

Summary:This opinion piece describes how the authors' ongoing field trial of genome-edited Camelina sativa plants changed legal status following a decision made by the European Court of Justice. Essentially, in a single day, the plants switched from not being regulated by the EU's GM directive to becoming just that. The article gives a personalised response to that decision; it also touches on the rationality of the ruling and its profound effects in the evaluation of crops in Europe.This is a story worth telling, but the following revisions would strengthen the manuscript.Essential revisions:-Refine the aim and take-home message1) While sympathizing with the authors' frustrations, the reviewers would encourage the authors to think more about what it is they hope to achieve with their article and make that clearer in the text. The article touches on many topics but there is often little in the way of facts that would persuade the readers of the importance of the issues.

We have made several edits to the text which hopefully address these concerns and make the article "punchier". We have now strengthened our messages in several points, especially towards the end of the text.

2) The reviewers were also puzzled what the take-home message is in the last paragraph. They all agree that field trials are essential to assess a plant's performance, but there was a feeling that this final section seemed slightly self-contradictory in places (for example, "Limiting the feasibility of field trials of GE by the complexity of the regulatory process.… will certainly hinder research" and "And whilst the application for Consent involves a detailed submission, this is no more arduous than that required for a manuscript for publication"). Please consider rephrasing this section to clarify and strengthen its message.

We have removed the last sentence. We agree it sent a confusing message. The message in subsection "The importance of field trials" is hopefully clear and unambiguous.

-Introduce more science and context3) For an unfamiliar reader, it is not clear what advantages genome editing has over conventional mutation techniques, since the authors state that the two approaches can do the same thing. The techniques should be more clearly outlined and the differences discussed.

We have amended the text in the Introduction and subsection "The making of a mutant" to help make this clearer.

4) Information should be provided on the biosynthesis of unsaturated fatty acids and the role of FAD2. This could be added in a Box, like https://elifesciences.org/articles/39233#box1, to avoid disrupting the flow of the main text.

We decided to modify the text in the Introduction to clarify the role of FAD2 in fatty acid biosynthesis. This seemed the simplest way to inform the reader.

5) It is clear to those working in the field that the EU's GM directive is directed to regulate the technology used in altering the properties of a plant and not the introduced property per se. Therefore, the argument that the authors put forward that traditional mutation breeding and mutations through CRISPR-Cas9 cannot be distinguished is correct, but this does not change the interpretation of the directive. The reviewers felt that the authors could discuss the problems with the GM directive concerning mutation breeding more to make these issues apparent to unfamiliar readers.

We have added a sentence to this effect in subsection "The making of a mutant" to address this issue.

6) The reviewers noted that in most cases CRISPR-Cas9 mutations through transgenes produce many off-target mutations and DNA rearrangements. Often pieces of DNA of the vector plasmid are integrated in the genome (which are not detected by PCR of the marker gene or CRISPR-Cas9 genes). Although off-target mutations are much more frequent in plants that have undergone conventional mutation breeding, it cannot be excluded that the authors' Camelina contain small pieces of foreign DNA. The authors are advised not to state that their Camelina is free from foreign DNA, unless it has been rigorously proved via whole genome sequencing.

As noted in the text, DEFRA (the UK competent agency) specifically did not ask for WGS data as to the GE plants contained any foreign DNA, but instead only asked for PCR confirmation of the absence of the transgene. But we agree our text was ambiguous on this point, so we have now edited it to make thigs clearer (subsection "Status update").

7) The authors should also be aware that, in 2017-2018, other field trials with genome-edited plants have been performed (including a trial with maize in Belgium). This and any other examples should be mentioned in the article.

To our knowledge, our trial was the first in which the scientists publicly sought regulatory clarification prior to sowing. We have now mentioned the VIB maize trial, which although approved by the Belgium competent authority, was run as a covert activity. We have contrasted the two approaches (Introduction).